# Improvement of the Electronic—Neuronal Interface by Natural Deposition of ECM

**DOI:** 10.3390/ma14061378

**Published:** 2021-03-12

**Authors:** Tobias Weigel, Julian Brennecke, Jan Hansmann

**Affiliations:** 1Translational Center for Regenerative Therapies, Fraunhofer Institute for Silicate Research ISC, 97082 Wuerzburg, Germany; info@isc.fraunhofer.de; 2Department Tissue Engineering and Regenerative Medicine, University Hospital Wuerzburg, 97070 Wuerzburg, Germany; lterm-assistenz@uni-wuerzburg.de

**Keywords:** neuronal electrodes, carbon fiber, electrospinning, ECM coating

## Abstract

The foreign body reaction to neuronal electrode implants limits potential applications as well as the therapeutic period. Developments in the basic electrode design might improve the tissue compatibility and thereby reduce the foreign body reaction. In this work, the approach of embedding 3D carbon nanofiber electrodes in extracellular matrix (ECM) synthesized by human fibroblasts for a compatible connection to neuronal cells was investigated. Porous electrode material was manufactured by solution coelectrospinning of polyacrylonitrile and polyamide as a fibrous porogen. Moreover, NaCl represented an additional particulate porogen. To achieve the required conductivity for an electrical interface, meshes were carbonized. Through the application of two different porogens, the electrodes’ flexibility and porosity was improved. Human dermal fibroblasts were cultured on the electrode surface for ECM generation and removed afterwards. Scanning electron microscopy imaging revealed a nano fibrous ECM network covering the carbon fibers. The collagen amount of the ECM coating was quantified by hydroxyproline-assays. The modification with the natural protein coating on the electrode functionality resulted in a minor increase of the electrical capacity, which slightly improved the already outstanding electrical interface properties. Increased cell numbers of SH-SY5Y cell line on ECM-modified electrodes demonstrated an improved cell adhesion. During cell differentiation, the natural ECM enhanced the formation of neurites regarding length and branching. The conducted experiments indicated the prevention of direct cell-electrode contacts by the modification, which might help to shield temporary the electrode from immunological cells to reduce the foreign body reaction and improve the electrodes’ tissue integration.

## 1. Introduction

The application of actuators or sensors in nervous or neuronal tissues requires consistent conditions concerning the electronic-tissue interface, to ensure stable electronic interactions between implant and neurons. However, the inflammatory reaction caused by the foreign body induces a highly volatile environment at the electronic-tissue interface. The attraction of immunological cells, changes in the local pH-value and the developing fibrosis reduces the sensitivity of the connection between electronics and electroactive cells in short and long-term applications [1]. The initial injury caused by the surgical implantation results in bleeding and tissue damage. In response to the injury, alarm signals such as intracellular proteins and matrix proteins activate the infiltration of immune cells to the implantation site. During this process, several immunological cell types like microglia, astrocytes or macrophages interact with the foreign structure and secrete soluble mediators like cytokines and proteases to the surrounding tissue. Concerning the cellular activation state, these cells can polarize to proinflammatory or anti-inflammatory states and thereby specify the extent of inflammation. This initial process remains for several days and briefly suppresses the electrodes performance. Several weeks later, the immune cells create a nonpermeable barrier between implant and tissue and recruit fibroblasts. Finally, the formation of a glial scar and the encapsulation continuously reduces the functions of the electrode implant [2].

In peripheral nervous system applications, the electrodes’ performance can be adjusted by regulation of encapsulation level and replaced after months or a few years [3]. Permanent applications, like implants for stimulation in the central nervous system for motor functions [4] or concepts of permanent brain–computer interfaces, require highly tissue compatible electrode surfaces with ideally no inflammatory reaction. This possibility might be enabled by electrode integration into the neuronal tissue supported by regenerative processes. Main requirements on neuronal implants are the biocompatibility, biomimicry and biostability [2]. Due to material improvements in recent decades, the compatibility and stability of the electrode material are minor issues. New developments focus on mimicing the electrode or the electrode surface to characteristics of extracellular matrix or the structure of neuronal cells [5].The objective of these improvements is to improve the neurite growth to the electrode and to avoid the activation and recruitment of glial cells as well as fibroblast, representing the main contributors to the capsule formation [2]. The major causes of inflammatory reactions of current electrodes are the material and its morphology. As differences in elasticity between implant and tissue triggers inflammation [6], conductive and stiff electrode materials like bare metals, are of no concern for future developments in neuronal implants. Thereby, the high Young’s modulus of metals like platinum or titanium nitride as well as the low flexibility in lots of electrode designs unavoidably provoke the rejection reaction by the cellular interaction with the electrode surfaces. Electrode flexibility can be achieved by generating thin metal layers on flexible support structures like Polydimethylsiloxan [7]. However, within a short time frame, the flat or also nanostructured surface is still recognized as a foreign body by the immune system. The electrode design as well as the materials need to be completely reworked to enhance the functional lifetime of neuronal electrode implants.

A possible improvement might be implants, which induce the interaction and ingrowth of whole cellular populations or at least cellular parts into the electrode material [7]. A strategy for cellular ingrowth aiming for inflammation reduction in soft tissues is the generation of an interconnected pore network with pore-sizes in the double-digit micrometer range [8]. Additionally, the implementation of flexible electrode designs may reduce damages caused by acceleration or other movement forces. New developments describe the bioinspired neuron-like electronics, which mimic the subcellular structural and mechanical properties by 3D nanofiber-like structures achieving a sustainably stable interface for at least three months in vivo [5].

However, a fundamental problem is that all available conductive materials still differ in their chemical surface composition to neuronal tissue, wherefore the durable prevention of the foreign material recognition by the immune system is hardly possible. Therefore, new strategies focus on shielding the electrodes surface with more tissue-like materials in order to prevent the direct electrode-tissue contact yet allowing electrical transmission. Examples are to enclose the electrode in the living target tissue like published for cardiac applications [9] or to develop synthetic neuronal compatible hydrogels as electrode coatings [10]. Moreover, coatings of extra cellular matrix improve the immunological compatibility to neuronal tissue [11,12]; however, the natural composition and structure cannot be generated artificially without much effort and thereby cannot completely shield against the immune system. To generate coatings of natural electrodes in extracellular matrix (ECM) on surfaces, cellular ECM synthesis is a promising approach. Thereby, ECM secreting cells are cultured on the desired surfaces and chemically stimulated to produce ECM proteins. After deposition of matrix proteins, the cells are removed and the cell specific biologically defined ECM structure remains on the surface [13]. This technique is an auspicious strategy to generate highly neuronal tissue compatible electrode coatings.

To address all mentioned requirements on future neuronal electrodes, we first generated highly porous carbon nanofiber scaffolds as electrical conducting electrode prestructures. The nano fibrous electrode approach reduced the average elasticity, which is hardly possible for metal-based electrodes. Additionally, the generated pores in the nanofiber electrode provided the electrode a high flexibility [14]. In the second step, fibroblasts were seeded on the carbon fiber electrodes to synthesize ECM proteins. After removing the cells, the electrical and biological performance of the modified carbon nanofiber electrodes was characterized. To test the suitability of the electrodes to serve as a substrate for neuronal tissue, SH-SY5Y cells were cultured on the ECM-covered structures. On the ECM-covered electrodes, the neuronal cell line showed significantly improved cell adhesion and neurite formation compared to standard surfaces.

## 2. Materials and Methods

### 2.1. Ethical Clearance Statement

Primary human dermal fibroblasts (hdf) used in this study were isolated from foreskin biopsies obtained from donors aged between 1 and 3 years under informed and written consent according to ethical approval granted by the institutional ethics committee of the Julius-Maximilians-University Wuerzburg (vote 280/18sc and 182/10).

### 2.2. Carbon Fiber Electrode Generation

Carbon nanofibers were generated by electrospinning a 12% polyacrlonitrile (PAN)/N,N-dimethylformamide (DMF) solution (Sigma-Aldrich, Schnelldorf, Germany) by applying 7 to 10 kV at a distance of 10 cm to a rotating target (diameter = 33 cm). With increasing scaffold thickness, the particular porogen destabilized the applied electrical field. To overcome this effect, the electrical field needed to be adjusted in steps from 7 kV at the beginning, up to 10 kV at the end of the spinning process. Afterwards, the PAN fiber mat was stabilized in a tube furnace (L08/14; Nabertherm, Lilienthal, Germany) at standard air atmosphere conditions with a heating rate of 2.5 °C/min to a final temperature of 250 °C for 2 h. The following carbonization was conducted in a tube furnace (R 50/250/12, Nabertherm, Lilienthal, Germany) with an argon atmosphere and a heating rate of 8 °C/min to a final temperature of 1000 °C for 1 h. To enhance the mesh openings, nano fibrous porogens were placed between the fibers. Thereby, polyamide 6 (PA6) (Sigma-Aldrich, Schnelldorf, Germany) as a second nano fibrous material was spun simultaneously with a concentration of 12% in 1,1,1,3,3,3-hexafluor-2-propanol (HFP). To prevent melting of the porogen during the stabilization, the final temperature was lowered to 220 °C and time increased to 4 h. During the carbonization, the PA6 fibers decomposed into volatile substances. During the spinning process and in order to enhance the pores integration, NaCl porogens with a maximum diameter of 80 µm were placed between the fibers during the spinning process. Therefore, NaCl-Particles (Sigma-Aldrich, Schnelldorf, Germany) were strewed in-between the nanofibers as an alternating process. Briefly, the spinning process was stopped periodically after 6 min, while the scaffold surface was wetted by ethanol and powdered with the particles. During carbonization by sublimation these particles were later removed.

### 2.3. SEM Imaging and Stuctural Analyis

Due to the excellent electrical conductivity, the carbon nanofiber scaffolds were imaged in a scanning electron microscope (SEM) (Supra 25, Carl Zeiss AG, Oberkochen, Germany) without any preparation. To measure and count the mesh openings, a surface area of 0.02 mm^2^ was imaged and analyzed by ImageJ (FIJI) (Version 1.52p, 2020, NIH, Bethesda, MD, USA). Thereby the threshold of the picture was set to get a black and white picture separating fibers and mesh openings. The plane of the mesh openings was measured afterwards with function “analyze particles”.

For imaging the cell or ECM modified electrodes, the samples were first fixed in 4% PFA (ROTI^®^ Histofix 4%, Carl Roth GmbH, Karlsruhe, Germany). The water was removed by passing the samples through an ascending ethanol-series, in detail: 30%, 50%, 70%, 100% ethanol (Carl Roth GmbH) for 30 min, respectively. Afterwards, the ethanol was replaced with 1,1,1,3,3,3-Hexamethyldisilazane (HMDS) (Sigma-Aldrich, Schnelldorf, Germany) in mixtures of 33%, 66% and 100% and incubation times of 30 min, respectively. Finally, HMDS was replaced again with 100% HMDS and the samples slowly dried under a fume hood. Before SEM imaging, the dried samples were coated with a 1 nm layer of platinum in a sputter coater (EM ACE600, LEICA, Wetzlar, Germany).

### 2.4. ECM Modification

Scaffolds were cut, clamped into 11 mm diameter cell crowns and afterwards sterilized by autoclavation. Hdf were seeded on top in a concentration of 30,000 cells per square centimeter. Cell culture medium consisted of dulbecco’s modified eagle medium (DMEM, Gibco^®^ Life Technologies, Carlsbad, CA, USA), containing 10% Fetal Calf Serum (FCS, LOT: BS 210601.5, Bio and SELL GmbH, Feucht, Germany), 1% penicillin/streptomycin (pen/strep, PAA, Coelbe, Germany) and either 100 or 500 µmol L-Ascorbic acid 2-phosphate sesquimagnesium salt hydrate (Asc) (Sigma-Aldrich, Schnelldorf, Germany) for proliferation and ECM synthesis stimulation. Medium was renewed three times a week. Cultivation lasted four weeks under standard cell culture conditions (37 °C, with an amount of 5% CO_2_ and a relative humidity of 95%). The complete decellularization process was conducted under sterile conditions. Therefore, specimens were rinsed thrice with phosphate buffered saline (PBS^-^) (Sigma-Aldrich) and incubated in decellularization solution containing 22.5 g/L sodium desoxycholate (Sigma-Aldrich, Schnelldorf, Germany) in deionized (DI) water. The procedure was repeated on the following day. Specimens were stored in pen/strep containing PBS- until further usage.

### 2.5. Immunofluorecent Staining and Imaging

Standard laboratory procedures were used for immunofluorescence (IF) staining of the cryo sectioned samples. Hdf were stained against Vimentin 1:1000 (Mouse, Abcam, Cambidge, United Kingdom). The ECM was stained against Collagen I 1:1000 (Rabbit, Acris Antibodies, Hiddenhausen, Germany, BP8028) and the cell line SH-SY5Y against β-Tubulin III (Rabbit, abcam, Cambridge, United Kingdom, ab78078). The applied secondary antibodies were donkey anti-Rabbit lgG, Alexa Fluor 555 and Donkey anti-Mouse lgG, Alexa Fluor 555 (both Thermo Fisher Scientific, Dreieich, Germany). Fluorescence images were captured with a fluorescence microscope BZ-9000 (Keyence, Neu-Isenburg, Germany) or a confocal microscope TCS SP8 (LEICA, Wetzlar, Germany). For confocal imaging, following laser wave lengths and detector rages were applied: 4′,6-Diamidin-2-phenylindol (DAPI): excitation at 405 nm and emission detection between 410 and470; Alexa Fluor 555: excitation at 561 nm and emission detection between 565 and 625 nm; confocal reflection to detect the nanofibers: excitation at 476 nm and reflection detection between 474 and 480 nm; autofluorescence detection of the ECM: excitation at 488 nm and emission detection between 491 and 545. The z-stack recording occurred with a distance of 1.5 µm between the single images in z direction.

Fluorescence microscopy images were analyzed by FIJI. The DAPI quantification was carried out on at least 20 pictures, captured by a 10× objective. The average intensity was normalized to the dense nonmodified electrode. Moreover, the neurite length was measured in the β-Tubulin III images by FIJI within a total area of 16.4 mm^2^.

### 2.6. Hydroxyproline Assay

In adaption to an essay from Reddy and Enwemeka [15] specimens were hydrolyzed in 2 mL reaction vessels with 6 M HCl (Carl Roth GmbH, Karlsruhe, Germany) for 20 h at 95 °C (Heating Block ThermoStat plus, Eppendorf, Hamburg, Germany). Due to the assay’s sensitivity regarding pH value, hydrolyzed samples were dried and dissolved again in DI water. Assay standard solutions in a range between 20 and 160 µg/mL were prepared dissolving L-hydroxy-proline (Sigma-Aldrich, Schnelldorf, Germany) in DI-water. For assay execution, 35 µL of each specimen and standard solution were transferred into a 96-well plate. 75 µL of chloramine-T reagent were added, which consisted of 1.27 g chloramine-T trihydrate (Sigma-Aldrich, Schnelldorf, Germany) in 20 mL n-propanol (Merck Millipore, Darmstadt, Germany), stocked up to 100 mL with acetate-citrate buffer per 100 mL solution. 75 µL of Ehrlich’s reagent (1.5 g 4-(dimethylamino)-benzaldehyde (Sigma-Aldrich, Schnelldorf, Germany) in 6.6 mL of n-propanol and 3.3 mL 70% perchloric acid (both Carl Roth GmbH, Karlsruhe, Germany) were added. Wells were sealed with Microseal^®^ ‘B’ PCR Plate Sealing Film (Bio-Rad, Watford, United Kingdom) followed by an incubation at 60 °C for 60 min. Coloration intensity measurement was performed by a photometer Infinite M200 PRO (Tecan, Crailsheim, Germany) at 570 nm.

### 2.7. Impedance Spectroscopy

8 × 12 mm^2^ pieces of the nanofiber electrode samples were electrically contacted and placed in a vessel with PBS^-^. Impedance spectroscopy was performed by a three-electrode setup containing a glassy-carbon counter electrode and an Ag/AgCl double junction reference electrode (both Metrohm Autolab, Utrecht, The Netherlands). Spectra were recorded in a frequency range from 0.1 to 10 kHz at an alternating sinusoidal voltage of 10 mV with a Potentiostat PGSTAT204 (Metrohm Autolab, Utrecht, The Netherlands). Data was recorded, calculated and fitted with the software Nova 2.1 (Version 2.1, 2016, Metrohm Autolab, Utrecht, The Netherlands).

### 2.8. Neuronal Cell Culture

Cell line SH-SY5Y (ATCC^®^ CRL-2266™, Manassas, VA, USA) was cultured on both dense and porous scaffold types in cell densities of 125,000 and 250,000 cells/cm^2^ for two weeks under standard cell culture conditions. Medium consisted of DMEM:F-12 (1:1, Gibco^®^ Life Technologies, Carlsbad, CA, USA), 10% FCS, 4 mmol/L L-glutamine (Sigma-Aldrich, Schnelldorf, Germany), 1% pen/strep (PAA) and freshly-dissolved 10 µmol/L retinoic acid (Sigma-Aldrich, Schnelldorf, Germany).

## 3. Results

To achieve a flexible and tissue-like electrode, carbon nanofiber scaffolds were chosen to form a basic electrode structure. Such fibers (Figure 1b) can be produced by electrospinning a polyacrlonitrile (PAN) solution (Figure 1a) and subsequent stabilization at 250 °C (air) and carbonization up to 1000 °C (argon) [16]. As dense nanofiber scaffolds did not allow cell migration into the scaffold, the later formed cellular ECM might be limited to the surface resulting weak binding forces between electrode and natural ECM coating. Therefore, a second material was spun simultaneously into the fiber scaffold as a space holder, which was removed later. This second material needed to be stable during the subsequent stabilization process as the structure of the PAN fibers was strengthened in this step. The melting of the porogen in this step would lead to a contraction by capillary forces resulting in the loss of the increased mesh openings. In this work, PA 6 was chosen due to its excellent electro spinning properties [17] and thermal stability. As the melting range of PA6 is below 250 °C, the stabilization temperature was lowered to 220 °C and time accordingly doubled to 4 h to ensure a sufficient stabilization of the PAN fibers. During the following carbonization, PA fibers disintegrated and vanished from the scaffold, resulting in clearly increased mesh openings (Figure 1c) compared to the dense carbon nanofibers (Figure 1b). The second disadvantage of dense carbon nanofiber scaffolds is their poor flexibility. By generating pores within the scaffold, the flexibility was enhanced in previous works [9,14]. Thereby, NaCl particles were positioned as porogens, which also vanished during the carbonization, between the fibers during the spinning process. In contrast to this previous work, the particle size was reduced from 80 to 125 µm to the sieve fraction under 80 µm. This approach reduced the lamellar morphology and thereby mechanical scaffold instabilities concerning delamination. Combining both porogens, NaCl particles and PA nanofibers, the mesh opening sizes again increased (Figure 1d,e) compared to the application of PA as single porogen (Figure 1c). The mesh openings’ size distribution of the different electrodes was measured by FIJI and collected in Figure 1f. The cross section of the developed highly porous carbon nanofiber electrode revealed a homogeneous distributed pore network (Figure 1f) with a very low lamellar character. The increased electrode flexibility was demonstrated by wrapping electrode stripes (7–9 mm width) around a cannula (d = 2.5 mm). While the dense electrode immediately broke during the first bending attempt, the porous electrode stripe could be wrapped completely (three times with the tested length) without any visible damage around the cannula (Figure 1h). Electrospun carbon nano fibers are often described with brittle behavior and are therefore unsuitable for many applications as the handling is difficile. The ability of bending the scaffold several times with a low bending radius demonstrated the significantly reduced probability of occurrence concerning a catastrophic material failure.

In the next step, the carbon fiber electrodes were modified by an ECM coating. To achieve an ECM coating, which is as natural as possible concerning composition, nanostructure, and biological integrity, tissue cells were seeded on the electrodes and stimulated to deposit a biological matrix coating. Therefore, hdf were seeded with a cell density of 30,000 cells/cm^2^ on both dense and porous electrodes and incubated under Asc supplementation for four weeks. During this incubation time, the cells were able to migrate through the complete porous electrode. SEM imaging revealed the formation of a dense cellular multilayer on the dense carbon fiber electrode (Figure 2a) after a four-week incubation. The porous electrode provided a high migration potential, wherefore the cells penetrated the scaffold and prevented a complete closure of the surface by the cells in places (Figure 2d). This effect was confirmed by staining the cross section of the cultured electrodes against vimentin. Thereby, cells were exclusively located on top of the dense electrode (Figure 2c left), while the porous electrode was populated across the whole scaffold thickness (Figure 2f left). After dissolving the cells with sodium desoxycholate-solution incubation, a protein-based nano fibrous network remained on both electrode scaffolds (Figure 2b,e). An IF-staining of the decellularized electrode cross sections confirmed a substantial presence of collagen I in the remaining extracellular protein matrix (Figure 2c,f right). Additionally, this staining demonstrated the same localization of the collagen I in and on the electrodes compared to the cell localizations (Figure 2c left, f left). This resulted in an ECM coating on top of the dense carbon fiber electrode and a more gradient-like ECM modification of the porous electrode. To overlay the fluorescence image with the inverted bright field image of the electrode scaffold, the contras was adjusted. Unmodified images of the immunofluorescence staining as well as the controls and the other channels are provided in the supplement, anti-vimentin staining Appendix A and anti-collagen I staining Appendix A.

The application of quantitative hydroxyproline assay indicated the amount of collagen on the electrodes by supplementing different amounts of Asc. Increasing the supplement from 100 µM to 500 µM also led to an increased hydroxyproline content (Figure 3a). Additionally, the 3D electrode led to an increased hydroxyproline amount compared to the dense electrode. The electrodes performances in PBS were tested by impedance spectroscopy. Thereby, the dense (Figure 3b) as well as the porous electrode (Figure 3c) showed a slight decrease in the amount of the impedance at low frequencies after ECM modification and decellularization.

Additionally, the phase shift was reduced by the ECM on both electrode types. The comparison between dense and porous electrode demonstrates only a slight shift in the phase maximum of around 3 Hz to lower frequencies. To analyze this behavior, the recorded curves were fitted on the electrical schematic in Figure 3d and gathered in Table 1. Differences between all samples, porous—dense—ECM-modified, were detected in the admittance of the constant phase element (CPE). As the ideality of the CPE stayed almost unaffected by the different samples, the electrical capacity was affected by electrode morphology and modification. Thereby, the dense electrode resulted in a higher capacity than the porous electrode. Additionally, the ECM modification resulted on both electrode types in a slight increase in the capacity, while R_CT_ stayed in all cases almost unaffected. The diffusion element T (or bounded Warburg element) [18] showed mainly differences between the porous and dense nanofiber electrode, as the scaffold thickness as well as the pore and mesh opening sizes suggest different diffusion behaviors. Thereby, the admittance Y_T_ was increased by the porous electrode and the time constant reduced.

The first step towards the application as a neuronal electrode requires the biocompatibility of the electrode with neuronal cells. Therefore, SH-SY5Y cells were seeded on the electrodes and differentiated for two weeks by the supplementation of retinoic acid. After IF staining against beta tubulin III, only small numbers of cells in form of small cell aggregates were found on the dense carbon fiber electrode (Figure 4a,b) indicating limited cell adhesion. The cell number was slightly increased on the porous electrode (Figure 4c,d), as the cells were trapped inside the electrode and protected from washing. This can be seen in Figure 4d, where black fiber shadows are above the cell cluster. For better visualization, 3D microscopy (Figure 4i) showed the cell cluster trapped below the first fiber layer.

By modifying the electrode with ECM, the cell adhesion was strongly improved resulting in an increased cell density. A DAPI intensity quantification showed comparable cell numbers on both unmodified electrodes, while the ECM coating increased the cell number by factor 2 on the dense electrode and factor 4 on the porous electrode (Figure 4k). Compared to the nonmodified porous electrode, the ECM coverage prevents infiltration of the cells (Figure 4j). Additionally to the cell adhesion, the ECM coating on the electrode boosted the formation of axons and their length. On the bare fiber electrodes, the neurites had an average length of 63.6 ± 43.2 µm for the dense and 84.6 ± 50.8 µm for the porous variation (Figure 4l). The ECM-modification increases the average neurite length on the dense electrode to 99.6 ± 58.9 µm an in the porous electrode to 120.2 ± 63.6 µm. In addition, the maximum neurite length increased up to 500 µm on the ECM-coated porous electrode compared to about 300 µm on the bare porous electrode.

By confocal microscopy, the axons could be visualized three-dimensionally in the porous electrodes. Thereby, the nonmodified porous electrode served as a 3D scaffold for the axons, which passed through the pores and extended mesh openings (Figure 5a) with hardly any branching. To achieve a better 3D-view, videos of the spinning pictures are provided in the supplement. By doubling the initially seeded cell number, the formed 3D axon network was more prominent (Figure 5c; Appendix A). In contrast, the modification of the porous scaffold with ECM avoided the penetration of the neuronal cells and their axons into the electrode (Figure 5b). The formed axon network was almost exclusively located on the electrodes’ surfaces and was not affected by an increased cell number (Figure 5d; Appendix A). Additionally, the ECM modification caused the formation of high numbers of fine branches on the electrode, which might improve cell–cell connections. To unhide the ECM on the electrodes without an additional staining, the autofluorescence of biological tissue in the GFP-excitation/emission-area was utilized. On the bare porous electrodes, just the cell bodies showed a slight fluorescence (Figure 5e). In contrast, the ECM covered porous electrode demonstrated a green fluorescence over the whole surface (Figure 5f; Appendix A). This visualization confirmed that the neurites were located on top of the ECM layer. Just the fine branches were able to penetrate the protein network in direction to the carbon fibers.

## 4. Discussion

To achieve permanent and functional neuronal tissue-electronic interfaces for sensitive signal transmission, the conventional electrode design needs to be redefined to entirely prevent inflammatory reactions on implanted electrode materials. New requirements for these electrodes are neuronal tissue adjusted mechanical properties like elasticity, flexibility, and material density [19]. Additionally, more cellular interaction between neurons and electrode is desirable to reduce the foreign body character with additional pores in the micrometer range [8,19] and fiber structures mimicking the neuronal cellular architecture [5]. Therefore, we developed modified carbon nanofiber scaffolds, which can improve the neuronal cellular interaction. Disadvantages of electrospun carbon nanofiber scaffolds are brittle mechanical properties and limited mesh openings sizes, which prevent any cellular infiltration. The generation of pores inside the nanofiber electrode highly increased the nanofiber scaffolds’ flexibility, as the thereby enlarged fiber distance enhanced their freedom of movement and thereby reduced the development of tension peaks during bending of the electrode scaffold. Compared to previous publications [9,14] the reduction of the particular porogen size reduced the delamination probability and improved the electrodes’ cohesion. By adding a second porogen in the form of nanoscaled electrospun fibers, homogeniously distributed wide mesh openings are ensured. The highly porous properties allowed the tissue cells to migrate through the scaffolds within four weeks vertically. In contrast, dense carbon nanofiber electrode limited the cellular growth to its surface, as it is usual for standard electrospun scaffolds [20]. After removing the cells, the electrodes were covered by a thin nano fibrous protein layer. This protein layer is morphologically comparable to ECM layers produced by fibroblast on polymer nanofiber by Mao et al. [13]. IF staining indicated that this network consists of collagen I to a large extent, which was localized at the initial positions of the hdf. The collagen’s cross-sectional observation revealed a gradient-like distribution from scaffold top to bottom with the highest amount at the cell seeding top site. It is assumed that this collagen distribution in the scaffolds interior leads to an improved adhesion of the ECM layer on the electrode, compared to the dense scaffold with a higher delamination probability. A previous study described the improved anchoring of collagen nanofiber layers by the combined spinning of collagen and PCL in a gradient-like structure [21]. Other ECM components were not characterized so far, but the presence of other collagen types or adhesion molecules like fibronectin and laminin is quite likely, as it was shown by other groups [13,22]. The ECM formation was quantified by a quantitative hydroxyproline assay, an amino acid with a content of about 13% in collagen I [23]. During the collagen synthesis, the amino acid hydroxyproline is formed by metabolizing Asc or other ascorbic acid derivates [24,25]. Testing of different Asc concentrations in the cell culture medium resulted in a higher hydroxyproline yield at a higher supplement. A concentration of 500 µM Asc is described as maximum before negative effects like pH shift reduce cell viability [25], wherefore we used aforementioned concentration as most effective supplementation. Additionally, the hydroxyproline formation was increased on the porous electrodes in each supplementation concentration by 30 to 50%. Due to the high porosity, the available proliferation surface is severely increased compared to the dense scaffold. This leads to higher cell populations in the porous scaffold, which produced more matrix in total, whereas the cell growth was slowed down by cellular contact inhibition. In the literature it is described that the collagen synthesis is increased in 2D culture compared to 3D culture [26]; however, in this case, the higher cell number in 3D might outbalance the higher collagen synthesis capacity in 2D. Regarding the electrical properties, the carbon nanofiber electrodes revealed excellent impedance properties, comparable to high performance TiN electrodes [27]. By fitting the measured data on the electrical circuit equivalent, the electrodes’ differences were characterized. The relatively low values of the charge transfer resistance R_CT_ and high values of the capacity C originates from the capacitive charge transfer [28] of carbonaceous materials. The reduced capacity of the porous electrodes can be explained by a lower active surface due to lower effective number of fibers compared to the dense electrode. By coating the electrodes with ECM, the capacity raised slightly in both cases. The thin protein layer increases the capacity by increasing the polarity on the electrode surface [29]. Instead of a classical Warburg element, fits were applied on a T element or bounded Warburg. At low frequencies, the diffusion of ions inside the electrode becomes more prominent. This effect can be described by the classical Warburg element not sufficiently. Regarding the diffusion of ions into the electrodes, the ECM coating showed just a marginal effect on the diffusion constants concluding, there are no negative effects on charge carrier permeability. In contrast, the different electrode types showed strong differences in charge carrier diffusion. Both electrode parameters are strongly influenced by the diffusion constant D [18]. A raised diffusion constant for an increased porosity results in a higher admittance Y_T_ and a decreased time constant B_T_. To note, B_T_ is proportional to the diffusion time of ions through the electrode.

In direct contact with the neuronal cell line SH-SY5Y, the modification clearly improved the cellular adhesion. Only proteins adsorbed from the applied cell culture medium supported the adhesion of cells on the bare carbon fibers. In contrast, the ECM modified electrodes provided large amounts of possible adhesion contact points. Additionally, the anchoring of these proteins in form of a matrix improves the resistance against cellular rinse off effects during medium change. The supply of a natural surface improved the differentiation by boosting the formation of axons, branching and the development of connections between the cell clusters. The higher cell number directly affects some of the mentioned improvements, but especially the formation of highly-branched nanoscaled axons originate due to the nano fibrous dense ECM network. The modified porous electrode demonstrated that the cells were not able to pass through this ECM coating. This behavior confirms initially the successful shielding of the synthetic electrode. However, currently it remains unclear whether more migration active immune cells are also blocked by the ECM layer, or if they are even stimulated to break down the coating by secreting enzymes [30]. However, the application of nonphysiological structured ECM coatings on electrodes could already improve the inflammatory properties in vitro [31] and in vivo [12]. The additional implementation of natural derived ECM structuring might improve previous electrode ECM-modifications.

Comparing the nonmodified scaffolds, higher cell number were found on the porous electrode. Here, the cells or cell aggregates were trapped inside the pores and reduced the rinse off effect, too. Due to this behavior, cells were able to differentiate inside the electrode and formed a 3D network of axons through the open pores and mesh openings. This may imply that the ECM coating can shield the electrode for a certain period from cell invasion and foreign body recognition. After remodeling or degradation of the protective ECM layer, the porous electrode can still act as a 3D scaffold and may stimulate the integration into the neuronal tissue due to its similarity to the neuronal architecture [5].

In conclusion, the extension of the electrospun carbon nanofiber electrode with two different porogen structures resulted in a porous scaffold with large mesh openings and high mechanical flexibility. The improved mechanical and structural properties enable the application as 3D electrode in the biological and medical field. Being covered with natural synthesized ECM by tissue cells provided an electrode surface with exceptional compatibility with the cell line SH-SY5Y. The imitation of ECM in composition and structure improved cell adhesion as well as neurite formation. Additionally, the ECM layer prevented the direct contact of the cell body to the electrode but allowed for penetration by small diameter neurites. The combination of a neuronal structure-mimicking electrode scaffold with physiological composed and structured ECM is a promising design for improved neuronal tissue integration. After the demonstration of the high compatibility with neuronal cells, the interaction with immunological cells is crucial for the success of this development. Therefore, the developed porous electrodes may positively influence the foreign body reaction, which needs to be further investigated.

## Figures and Tables

**Figure 1 materials-14-01378-f001:**
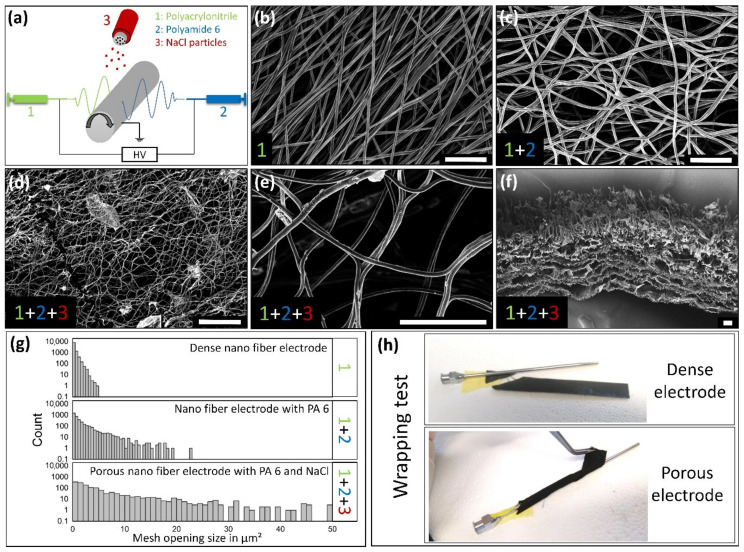
Electrode synthesis and structural properties. (**a**) Schematic E-spinning setup demonstrates the coaxial spinning process with simultaneous generation of two nano fibrous polymers, namely polyacrylonitrile (PAN) and polyamide 6 (PA). NaCl particles were sprinkled on the nanofiber mesh during short interruptions of the spinning process. (**b**) SEM image of a dense electrode from carbonized (PAN) fibers completely without porogens. (**c**) The addition of PA nanofibers in the spinning process results in an increase of the mesh openings. (**d**,**e**) Further integration of particular porogens (NaCl) lead to an additional increase of the mesh opening sizes. (**f**) Cross section reveals the homogeneous distributed pores formed by the NaCl particles. (**g**) Histogram of the mesh openings shows the quantitative increase caused by the addition of the porogens. (**h**) The loose fiber network as well as the pores greatly improves the nanofiber scaffolds’ flexibility, which was demonstrated by wrapping stipes of the carbon fiber scaffolds around a cannula (d = 2.5 mm). Scale bars correlate to 10 µm in (**b**,**c**,**e**) and to 50 µm in (**d**,**f**).

**Figure 2 materials-14-01378-f002:**
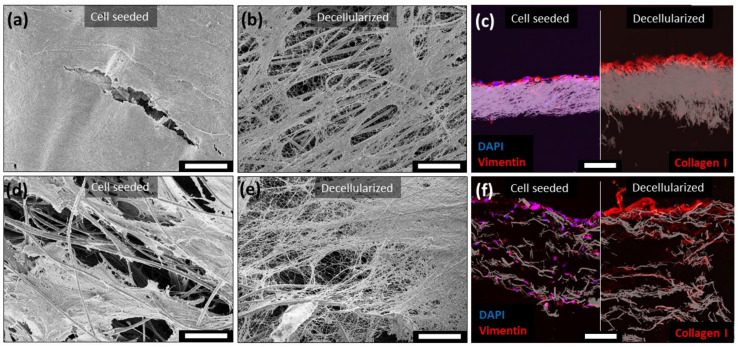
Electrodes in extracellular matrix (ECM) modification by human dermal fibroblasts (hdf). (**a**) SEM image of the sell-seeded surface demonstrates a closed cellular surface after four weeks of culture on the dense meshes. (**b**) After decellularization, a nano scaled protein network remained on top of the carbon fibers. (**c**) Immunofluorescence staining of the cross section affirmed the location of hdf (vimentin, left) and the ECM (Collagen I, right) just on top of the dense electrode. (**d**) Enhanced mesh openings and porosity improved the cell migration into the nanofiber electrode and reduced the thickness of the cellular layer on top of the scaffold. (**e**) Synthesized ECM on top of and in the porous electrode. (**f**) The porosity led to the colonization inside the electrode by hdf (vimentin, left) as well as the ECM synthesis (Collagen I, right). Scale bars correlate to 10 µm.

**Figure 3 materials-14-01378-f003:**
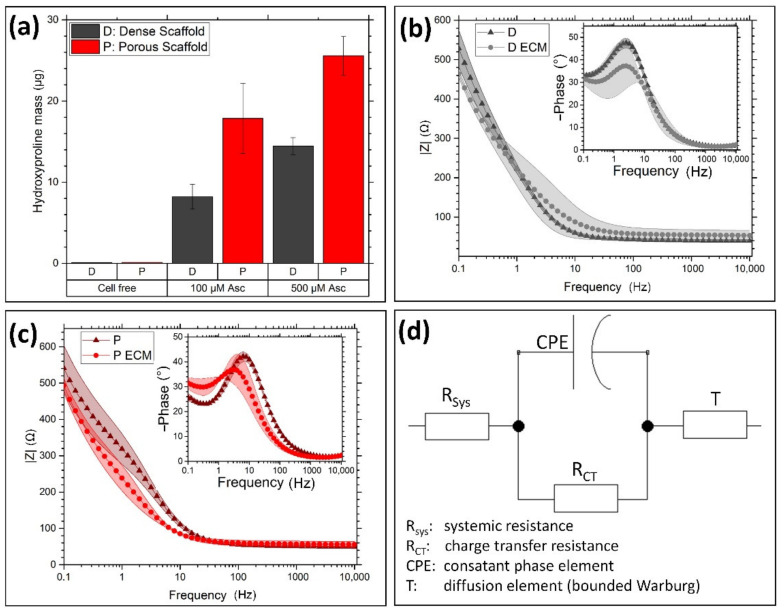
Electrode characterization. (**a**) The hydroxyproline assay showed proportional ECM formation by increased supplementation of L-Ascorbic acid 2-phosphate (Asc) as well as by increased electrode porosity (*n* = 3). (**b**) Impedance spectroscopy of the dense scaffold as well as (**c**) the porous electrode showed a slight decrease in the amount of the impedance. (**d**) The measured impedance spectra (*n* = 3) were fitted on the depicted electrical circuit to determine the single components.

**Figure 4 materials-14-01378-f004:**
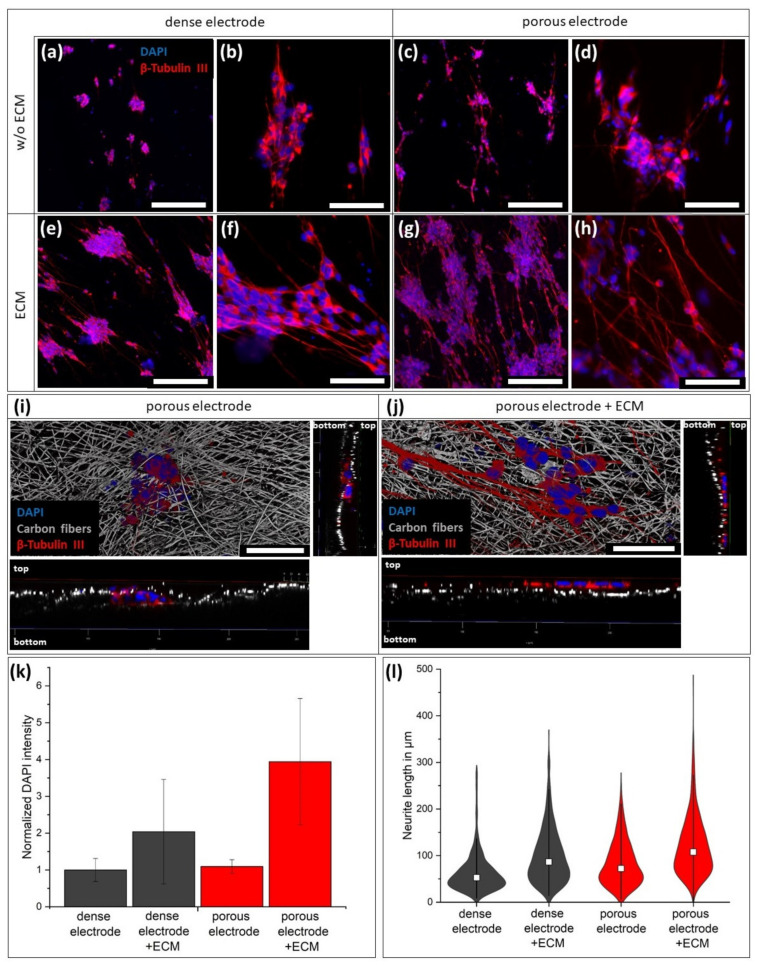
Behavior of SH-SY5Y cells on the ECM modified electrodes. (**a**–**d**) IF staining against β-tubulin III presented a reduced cell adhesion on the bare carbon nanofiber electrodes compared (**e**–**h**) to the ECM-modified electrodes. (**a**,**b**,**e**,**f**) describe the dense electrodes and (**c**,**d**,**g**,**h**) the porous electrode. Confocal imaging (confocal reflection imaging of the carbon fibers) showed the cells inside the bare porous electrode (**i**), while the ECM modification on the porous electrode prevented cellular penetration of the surface (**j**). Scale bars correlate to 200 µm in (**a**,**c**,**e**,**g**) and to 50 µm in (**b**,**d**,**f**,**h**–**j**). Quantification of the average DAPI intensity on at least 20 fluorescence images with subsequent normalization to bare dense carbon fiber electrode (**k**). Violin plot of measured neurite length of β-tubulin III-fluorescence image in a total area of 16.4 mm^2^ (**l**). White boxes represents the average neurite length and the breadth of the violin body is related to the number of measured neurites.

**Figure 5 materials-14-01378-f005:**
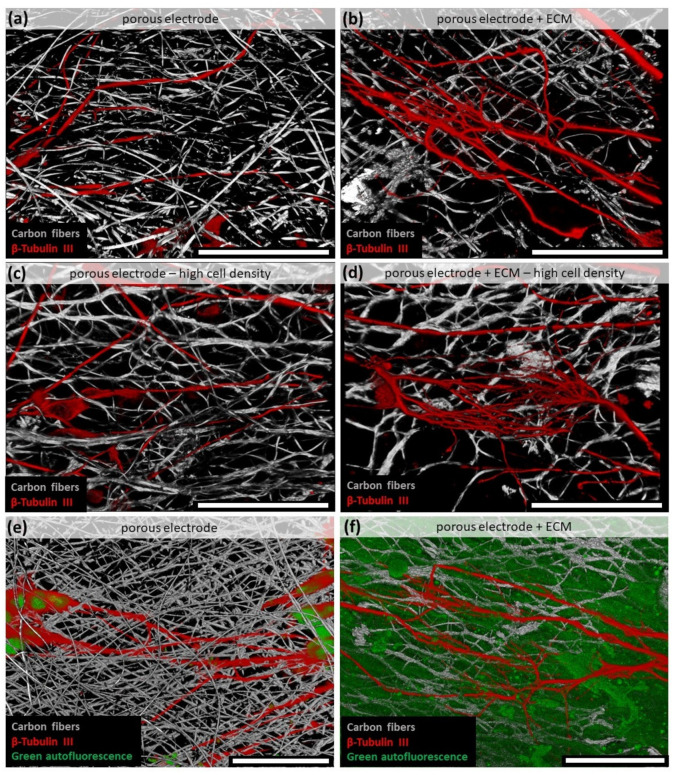
3D imaging of axons on the porous carbon nanofiber scaffolds. (**a**) The unmodified porous scaffold acts as 3D scaffold for the axons. (**b**) After ECM modification, the axons were more located on the electrode surface but severely increased in branching. (**c**,**d**) By raising the seeded cell number, the number of axons in/on the electrode increased. When addressing the autofluorescence of the biological tissue in the GFP related excitation/emission area, just the cell bodies on the bare carbon fiber electrode showed a green emission (**e**). The ECM modified porous electrode displayed a green fluorescence overall the electrode surface (**f**). Scale bars correlate to 50 µm in all images.

**Table 1 materials-14-01378-t001:** Resulting values of the impedance data on the electrical schematic in Figure 3d. Compared were the basic carbon nanofiber electrode, dense (D) and porous (P), as well as the ECM modified samples. R_Sys_ describes the systemic electrical resistance of the whole measurement setup; Y_CPE_ the admittance of the CPE; N_CPE_ the ideality of the CPE; R_CT_ the charge transfer resistance; Y_T_ the admittance of the diffusion element; and B_T_ the time constant of the diffusion element.

Electrode Type	R_Sys_ Ω	Y_CPE_ s^N^/mΩ	N_CPE_	R_CT_ Ω	Y_T_ s^1/2^/mΩ	B_T_ s^1/2^
D	40.1 ± 3.4	0.95 ± 0.06	0.83 ± 0.02	407 ± 42	77.9 ± 26.3	0.110 ± 0.028
D ECM	44.7 ± 3.1	1.13 ± 0.18	0.85 ± 0.01	337 ± 8	89.2 ± 9.6	0.103 ± 0.017
P	49.4 ± 1.7	0.42 ± 0.05	0.81 ± 0.01	387 ± 64	183.5 ± 59.5	0.046 ± 0.014
P ECM	51.2 ± 4.5	0.68 ± 0.01	0.81 ± 0.001	357 ± 8	111.0 ± 32.1	0.041 ± 0.012

## Data Availability

Data sharing is not applicable to this article.

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
