# Peer review of "Improvement of the Electronic—Neuronal Interface by Natural Deposition of ECM"

_materials, 2021, doi:10.3390/ma14061378_

Round 1

Reviewer 1 Report

1. The authors have stated that the issue with implantable electrodes is foreign body response. Such as response is an immunological response that involves interaction between macrophages and several other cell types. Irrespective of any material used, there is bound to be a immune response, even in the case of healing. The introduction does not feature any details about what aspect of foreign body response are harmful that poses an impediment to the success of implantable sensors. The authors may want to research into these aspects in detail especially if they want to deal with inflammation 2. Line 63: The references provided talk about the chronic stage but the authors are doing an acute study. How is this relevant? 3. Use of ECM coating on nanofibers: Is this novel? there have been many studies previous done. The novelty of the study is not clear. There are several products in market as well for in vitro culture. 4. Use of of ECM based decellularized scaffolds can also give rise to very very high inflammation and even immune rejection. The present study does not do any assessments to show that the scaffolds can reduce inflammation. 5. While the idea of the study is good, the design of experiments is not optimal. For examples, in order to show that such scaffolds will reduce inflammation, a macrophage culture or any in vivo test need to be done, based on what the authors have written in the introduction. If the goal is not to assess foreign body response, then the introduction needs to be changed. 6. The controls of immunofluorescence are missing? Can the authors show negative controls?It looks like there is a lot of background stain. 7. how was the 3D imaging done. Where are the controls ? The images are 2D images, can the authors show 3D images? 8. this paper does not have a conclusion section at all 9. The conclusion summary saya in line 415 that there is exceptional biocompatibility. However, there is no evidence? This is not related to the introduction which talks about foreign body response. These are 2 different events which need to be taken into consideration. 10. Authors may want to thorough check the English grammar. The sentence formations are incorrect in multiple places including the first line.

Reviewer 2 Report

The manuscript reads well and the electrode platform developed has potential in the neural interface field. However, the authors claim results that need a clear explanation with solid data to be able to bring this work up to high standard. This lack of data is specially observed in the biological interaction of the electrodes with the neuronal cell line SH-SY5Y.

Comments:

1.“ The supply of a natural surface improved the differentiation by boosting the axon formation, branching and the development of cell-cell contacts”.

How can we see this? Further analysis of the branching and cell morphology needs to be done (Neurite length, cell area, cell spreading). Immunostainings for focal points should be done in order to say “the development of cell-cell contacts”

  1. “The higher cell number directly affects some of the mentioned improvements, but especially the formation of highly-branched nano scaled axons can be traced back the nano fibrous dense ECM network. The modified porous electrode demonstrated that the cells were not able to pass through this ECM coating"

Can the authors show this in a 3D rendering of electrodes?

  1. "This behavior confirms initially the successful shielding effect around the synthetic electrode".

How? Can the authors elaborate on this? Further analysis or data to backup this observation?

  1. “After remodeling or degradation of the protective ECM layer, the porous electrode can still act as a 3D scaffold and may stimulate the integration into the neuronal tissue due to its similarity to the neuronal architecture"

Can the authors show this with data or further analysis of the morphology of the neurons culture on ECM electrodes and non ECM electrodes?

Reviewer 3 Report

The article titled “Improvement of the electronic – neuronal interface by natural deposition of ECM” by Weigel et al. on the synthesis of a nanocomposite material based on PAN/PA and ECM from human fibroblasts towards electronic – neuronal interface is an interesting work. I would suggest following changes be made in the manuscript:

  1. Significant grammatical corrections are needed in the manuscript, especially the abstract needs to be revised to include some of the results in terms of conductivity and statistical analysis on the increased number of neural cells suggesting biocompatibility.
  2. The manuscript needs a lot of work in terms of syntax and proper usage of the wording. For e.g. on the very first line of the Abstract, it reads “Due to still occurring foreign body reaction to neuronal electrodes implants…” which seems like the authors are trying to talk about contamination of electrode implants? First line in the Introduction section reads “The application of actuators or sensors in nerve or neuronal tissues require constant conditions in the tissue to achieve stable interactions…”, which I am not sure what the authors are trying to convey. Is it the implanted actuators and sensors or something else? Also, what constant conditions in the tissue? On line 28, the authors state that “the foreign body induces a highly dynamic environment at the implant area”, which again is too vague. There are many such descriptions that other than punctuation errors, are just too hard to perceive.
  3. On line 91, the authors stated that the applied voltage for the electrospinning process was 7 to 10 kV. Why this range? Were there multiple types of nanofibers synthesized? If so, which voltage provided the best nanofiber diameter, porosity, etc?
  4. Figures 3b and 3c should have different symbols/ colors to indicate differences between the impedance of porous and dense scaffolds. Also, the slight decrease in the impedance in case of porous scaffolds may be due to the presence of salt crystals. Is there any way the authors can convincingly show that the NaCl crystals were not present at the time of performing the EIS? May be EDAX?
  5. For figures 4 and 5, the authors should provide the statistical numbers for the cell count in case of the control, ECM and ECM modifier porous scaffolds to get a better understanding of the increased adhesion instead of imaging only.
  6. It is recommended that the authors either show some conductivity data based on the biological response to conclusively prove the positive interactions between the neural cells and scaffold or atleast insert a figure depicting the neurite growth over a period of time on the nanocomposite scaffold.
  7. Elasticity data on the nanocomposite scaffolds is also not clear, which the authors should describe in the results section more succinctly.
  8. The Conclusion section needs significant work where the authors should point out the pros and cons of the synthesized scaffold and its ability to stimulate neuronal adhesion.

Round 2

Reviewer 2 Report

The authors have addressed all comments.